# Social and Family Challenges of Having a Child Diagnosed with Phelan-McDermid Syndrome: A Qualitative Study of Parents’ Experiences

**DOI:** 10.3390/ijerph191710524

**Published:** 2022-08-24

**Authors:** Cristina García-Bravo, Domingo Palacios-Ceña, Sara García-Bravo, Jorge Pérez-Corrales, Marta Pérez-de-Heredia-Torres, Rosa Mª Martínez-Piédrola

**Affiliations:** 1Research Group in Evaluation and Assessment of Capacity, Functionality and Disability of Universidad Rey Juan Carlos (TO+IDI), Department of Physical Therapy, Occupational Therapy, Physical Medicine and Rehabilitation, Universidad Rey Juan Carlos, 28922 Alcorcón, Spain; 2Research Group of Humanities and Qualitative Research in Health Science of Universidad Rey Juan Carlos (Hum&QRinHS), Department of Physical Therapy, Occupational Therapy, Physical Medicine and Rehabilitation, Universidad Rey Juan Carlos, 28922 Alcorcón, Spain; 3Physiocare Madrid, Physiotherapy Clinic, 28026 Madrid, Spain

**Keywords:** Phelan-McDermid Syndrome, rare diseases, parents, qualitative research, social environment, social support

## Abstract

(1) Background: Phelan-McDermid Syndrome (PMS) in children causes significant challenges affecting social and family relationships. The purpose of this study was to explore the experience of parents with children diagnosed with PMS regarding interactions with their social environment; (2) Methods: A qualitative descriptive study was conducted. Participants were recruited using non-probabilistic purposeful sampling. In total, 32 parents of children with PMS were included. In-depth interviews and researchers’ field notes were used to collect the data. An inductive thematic analysis was performed; (3) Results: Five themes were identified: (a) challenges in the relationship as a couple; (b) challenges within the family and close social relationships; (c) challenges in the educational-school environment; (d) challenges in the health environment and with health professionals, and (e) reconnection through the PMS association. It would be beneficial for parents to create training programs on PMS in the educational and healthcare settings, to promote the participation of professionals in the PMS association and to develop care programs focusing in their physical, psychological and social health.

## 1. Introduction

Phelan-McDermid Syndrome (PMS) is a genetic rare disease (RD) that is associated with severe cognitive impairments such as intellectual disability, absent or delayed speech, and symptoms of autism spectrum disorder (ASD) [1], along with disruptive behaviors [2,3]. Moreover, skill losses occur in adolescence, and neuropsychiatric symptoms such as bipolar disorder, catatonia, psychosis, language loss and sudden behavioral changes may appear [4,5].

As a result of the symptoms of the disease and behavioral alterations, the parents’ and family’s social relationships are negatively impacted [2,3]. Parents of children with other RD have highlighted the constant concern for their children and people’s lack of understanding [6,7,8]. Parents with children diagnosed with RD face numerous difficulties including diagnostic confirmation, daily care of the children, and symptom management [9]. This causes direct and intense individual and social involvement of parents [9].

Previous studies [10,11] among parents of children with RD showed how their health status worsens when they experience limitations for participating in social relationships and feelings of isolation and loneliness [10,11]. Such social limitations are evident in the relationship with family members due to the burden of care and the stress they experience [10,12,13], in the relationship with friends due to lack of time [10], in the relationship with healthcare professionals due to lack of understanding, coordination and awareness of the disease [10,14], and in relationships with the children’s educational staff due to lack of understanding of the pathology [12].

Special attention is given to the interaction with the educational environment in schools. School care for children with special educational needs was regulated in Spain in 1995, and the parents’ free choice between a standard school or a special education center was established [15]. However, the schools themselves are those that assess and decide whether they have the capacity to be able to provide quality educational care to children with PMS. In the event of not being able to provide adequate care, the schools refer the child to other centers that are adapted to meet the needs of the child with PMS [15].

The experience of having children diagnosed with PMS or an RD is highly individual, and qualitative research can provide a more holistic view that may be more meaningful to parents [16]. In contrast, there are no previous qualitative studies describing the experience of parents of children diagnosed with PMS. The objective of this study was, therefore, to explore the experience of parents of children diagnosed with PMS in terms of interaction with their social environment.

## 2. Materials and Methods

### 2.1. Study Design

A qualitative descriptive exploratory study was conducted [17,18,19,20]. The theoretical framework that guided this study was interpretivist [21]. With the use of qualitative methodology, the aim was to understand the beliefs, values, and motivations that underlie individual health behaviors [22]. The purpose of an explorative descriptive qualitative study is to identify an event or a critical situation. It seeks to show “what is happening” and “how it is happening” [19,20]. Qualitative exploratory studies aim for a comprehensive summary of events in the everyday terms of the described event. This design is the method of choice when straight descriptions of phenomena are desired [17,18]. Moreover, from an interpretive perspective, human action is meaningful, and the goal of inquiry is understanding how people respond to and understand the meaning of social phenomena [21,23]. This study was conducted according to the Standards for Reporting Qualitative Research (SRQR) [24] and the Consolidated Criteria for Reporting Qualitative Research (COREQ).

The study was approved by the Local Ethical Committee of Universidad Rey Juan Carlos (code: 0810202017820). All participants provided oral informed consent prior to their inclusion [21]. Six researchers (two men) participated in this study, including two research nurses (DPC, SGB), and four occupational therapists. All researchers had experience in research in health sciences. The study protocol was audited by an external researcher.

### 2.2. Participants, Context, and Sampling Strategies

The study included parents with children diagnosed with PMS by medical specialists and who were attending the Asociación Síndrome Phelan McDermid (https://22q13.org.es/, accessed on 15 June 2022), Madrid (Spain). The inclusion criteria accounted for the following. (a) Parents who, at the time of the study had children diagnosed with PMS (with genetic diagnosis), and/or the legal guardian; (b) the diagnosis of PMS was made by the pediatrician and/or the neurologist, (c) children could present any variation of PMS (deletion or mutation), and (d) signing informed consent. Exclusion criteria included a diagnosis not confirmed by the pediatrician and/or neurologist, and not signing the informed consent.

Purposive sampling was used, based on relevance to the research question (not clinical representativeness) [21]. In the current study, the sample size was determined following the Turner-Bowker et al. proposal [25]. These authors [25] reported that 99.3% of concepts, themes, and contents emerged with around 30 interviews. With this proposal, a greater capacity to identify codes, categories, and topics is achieved. The current proposal also helps researchers to know when to stop collecting data and recruiting participants.

### 2.3. Data Collection

Data were collected over a four-month period between November 2020 and February 2021. In-depth interviews were used as the main data collection tool. During the interviews, a semi-structured question guide was used (Table 1). Moreover, researcher field notes were kept during the interviews [26]. During the interviews, researchers used prompts to encourage the participant to provide further details.

The interviews were conducted via a private video chat room using the Microsoft Teams platform (https://www.microsoft.com/es-es/microsoft-teams/log-in, accessed on 15 June 2022). Each participant received a private and personalized email with an invitation. All interviews were conducted by three authors (CGB, RMMP and DPC). After obtaining oral permission from participants, all interviews (*n* = 32) were audio-recorded and video-recorded, recording a total of 3205 min of interviews overall (average of 100.16 ± 18.2 min each interview).

Additionally, during the semi-structured interviews, 32 field notes were collected by the researchers. Researchers’ field notes provide a rich source of information concerning how participants describe their personal experiences, as well as participants’ behaviors during data collection, and they enable researchers to describe their reflections concerning methodological aspects of the data collection [26]. See Table 2.

### 2.4. Data Analysis

The full verbatim transcript of each of the interviews, and the researchers’ field notes, were collated to perform a qualitative analysis [21,26]. An inductive thematic analysis was performed [21]. The analysis consisted of identifying the most descriptive content in order to obtain codes and subsequently reduce and identify the most common meaningful groups (categories). In this manner, groups of meaningful units were formed (i.e., similar points or content that allowed the emergence of the topics that described the study participants’ experience) [21,26]. This process was performed separately on the interviews and the researchers’ field notes. Double and independent coding was performed by two investigators in each interview and each field note (CGB, DPC). Subsequently, they met to discuss, compare, and refine their findings. Comparation and refinement processes included study and evaluation of category definitions, and characteristics and narratives which justified all categories obtained. After that, the same process was conducted with themes. Additionally, joint meetings were held to combine the results of the analysis, and to represent the parents’ experiences [21,26]. In the case of differences in opinion, theme identification was performed based on establishing a consensus. Figure 1 shows the coding process. No qualitative software was used to analyze the data. For the analysis, the Excel program was used to organize and share the coding process. Excel allows making grids, with hyperlinks that allowed us to organize all the codes based on their categories, as well as to associate them (link them) to the fragments of texts from the interviews that justify their existence. Finally, the external researcher, who audited the study protocol, checked the description of the coding tree, the major themes, participants’ quotations, quotation identification, and theme descriptions.

### 2.5. Rigor

The techniques performed to control trustworthiness are described in Table 3 [21,22].

## 3. Results and Discussions

Thirty-two parents were recruited (23 women). The mean age of parents was 43.38 years (SD ± 6.85). The mean age of children with PMS was 11.66 years (SD ± 9.79) and the child’s age at diagnosis was 7.04 years (SD ± 10.04) with a mean evolution of the diagnosis of 4.62 years (SD ± 4.88).

Five specific themes emerged describing challenges in the social relationships of parents at different levels: (a) challenges in the couple’s relationship; (b) challenges in the family and their relationship with their close social environment; (c) challenges with the educational-school environment; (d) challenges with the health environment and health professionals, and (e) reconnection through the PMS association. The results are presented with examples of narratives obtained from participant interviews. The use of narratives helps to maintain credibility and traceability of the results [22]. We also integrated the discussion with the results to contextualize the findings obtained.

### 3.1. Theme 1: Challenges in the Relationship as a Couple

All the participants described how the disease, the process of diagnosis and daily care are continuous challenges in life as a couple. The couple must make decisions related to the care of their child, and sometimes they cannot reach a consensus, leading to arguments and/or reproaches. Depending on the participants, the disease can either strengthen the couple or break it up:


*“I think these things either bring you together or completely separate you. In my case, it brought me much closer to my wife. Without her, I couldn’t have gotten through the diagnosis.”*
(P32).

Sometimes, however, the couple breaks up were due to the non-acceptance of the disease and constant arguments:


*“This affects you to such an extent that the situation is no longer sustainable. The way we treated each other changed. We didn’t talk, we didn’t understand each other… we just argued. And we decided to separate.”*
(P19).

The key to overcoming arguments, and strengthening the couple is communication, “team building”, respect, patience and admitting that all members of the couple suffer:


*“Your marriage contract didn’t include this extra work and these difficulties… but in the end, you need to have a lot of patience, a lot of respect for the other person… realizing that the other person is going through the same things as you… and is suffering the same, and talking about it, letting it all out…”*
(P17).

Most participants reported that sexual relations between the couple had either disappeared or become greatly reduced due to fatigue, the couple’s problems, and involvement in the ongoing care of their child:


*“We are almost roommates with burdens. No sexual relations or anything. No energy or desire to do anything.”*
(P24).

Many of the participants narrated how this situation made them feel bad. Several parents considered sexual relations as a fundamental aspect of the couple that they must attend to, because it is a necessary part of their relationship in order to be able to continue:


*“It really affects us and even though we don’t have time or strength, we always try to do something together, so that this part of the couple doesn’t die because I think it’s very important.”*
(P6).

The care required by children with PMS requires couples to invest daily time, which can lead to social isolation, emotional problems, and role changes [7,16,27]. However, there are no previous qualitative studies describing its impact on the couple. Our results show how sexuality is compromised, as reported by the participants. This aspect should be considered for future research.

Moreover, relationships within the family may be affected, with feelings of isolation, lack of support/understanding, a sense of being overwhelmed, and periodic family crises [28,29,30]. Witt et al. [28] describe how the implications of RD on family relationships needs to be addressed, being the fundamental basis of children’s care and development [28].

### 3.2. Theme 2: Challenges within the Family and Close Social Relationships

In many cases, parents reported that they stopped attending family gatherings (Christmas, birthdays, other celebrations) because they were misunderstood by their own family (looks, inappropriate questions, etc.), leading to feelings of discomfort and stress:


*“Relationships with the family are no longer the same… in the end you stop doing things with them because ultimately you get more stressed. The most important thing for everyone is their children’s problems and no one puts themselves in your place…the worst thing is the looks of pity…”*
(P3).

However, most parents, despite their estrangement, needed the support of their families on a daily basis to help them care for their children and make time for life as a couple and as individuals.

The parents also described how they stayed away from and decreased social relationships, stopped visiting friends, and decreased outings to leisure events. The main reason reported was the lack of understanding displayed by others regarding their situation and their child’s illness:


*“… in the end you don’t go out because it’s bad faces, they [people around you] don’t understand and it seems to bother them. I can’t control my little girl needing to scream or moan. Sometimes to go to suffer or have a drink in a hurry, rushed and overwhelmed and stressed about whether they [people] will say something, well I may as well not go.”*
(P8).

Despite losing some friendships, the parents pointed out how over time new friendships were created stronger than the previous ones:


*“We have met a lot of new people by getting into the disability world, and I think these relationships are richer and stronger. We understand each other because we experience very similar situations.”*
(P19).

Our participants highlighted the pain, sadness and suffering due to the rejection of children with PMS and the looks they receive:


*“It’s very hard when you see how people laugh at your son, or how they don’t understand him… they don’t understand him and they don’t understand us…”*
(P8).


*“… people stare very blatantly… even unkindly, and you notice that only people who are immersed in this world are the ones not staring at you… it’s very sad.”*
(P17).

Not only did parents experience incomprehension within the family, indeed, the siblings of the child with PMS are also affected:


*“… his siblings have had fights at school because he can’t see them picking on his sick brother, he takes it very badly and doesn’t understand why they laugh at him.”*
(P30).

Previous studies [6,29] among parents with children with RD described a disruption of the parents’ social relationships, due to the children’s challenging behaviors, physical fragility, fecal incontinence and/or the stigma that others have towards them. Sometimes, parents decided to self-isolate to avoid frustration or feeling forced by friends [6,29]. In agreement with these results, our findings show that the parents chose to social distance, because of the incomprehension among their environments. According to our participants, this incomprehension was expressed in the form of looks and gestures from the people around them, and complaints about their children’s behavior within establishments. This was a source of pain and anguish. Previous qualitative studies on parents of children with RD described the presence of perceived stigma and how it negatively influenced parents’ emotions and their participation in social and family relationships [6,29,30].

### 3.3. Theme 3: Challenges in the Educational-School Environment

Many parents pointed out that conventional schools failed to understand children with PMS, failing to meet their needs, and making them feel that their children were disruptive at school. In addition, parents highlighted the struggle they have at conventional schools to prevent their child from failing the academic year:


*“At school he’ s the kid who’s annoying. And who does he get stuck with every year? The one [teacher] who is new to the school. In the end, either you go and fight with the school and he’s the one who gets hurt, or you don’t do anything, and we see this year go by and next year we’ll see.”*
(P30).

Many of the difficulties that children with PMS encounter in the standard school are due to the manifestation of the disease (disruptive behavior, screaming, whining, lack of concentration) that disrupts the class and forces the teacher to focus on the child with PMS:


*“… they don’t understand that he doesn’t do it on purpose, and they told me he was spoiled because of me.”*
(P2).

For our participants, a special education school is essential for the proper development of their children and to achieve a balance between meeting their needs and the child’s education and training. Parents felt that the support of the special education school was crucial for their child and their educational process:


*“… She is very much supported at school, and they are very supportive of her. The support is both academic and human, and for me it has been crucial. I value that more than the fact that they learn.”*
(P14).

Regarding the presence of RD at school, previous studies [12,31] showed how the presence of an RD impairs the child’s ability to adapt to the school environment due to the presence of difficulties in speech, behavior and/or learning difficulties. Moreover, the lack of knowledge of the disease and the reluctance or refusal of school staff to work with the children, make social isolation of these children more likely [12,31].

The studies by Melin et al. [31], and Kuhn and Albertowski [32], show that most parents with children with learning difficulties, autism and/or RD go to schools that specialize in their children’s pathologies to meet their needs and continue with their education. Our participants highlighted the importance of support from specialized schools and having educational professionals trained to manage children with PMS. Previous studies described how an optimal collaboration between medical services, school and parents improved the quality of life of children with RD at school [32].

### 3.4. Theme 4: Challenges in the Healthcare Environment and with Healthcare Professionals

The main challenge faced by parents was finding doctors who are knowledgeable and experts in PMS. Most narrated feeling that physicians lacked the expertise to manage their children with PMS, due to lack of knowledge:


*“… in neurology they are supposed to be the ones with more specialization or the ones who give me more answers… but that’s not the case. What the neurologist does is ask me how the child is doing and gives me an appointment for a year from now.”*
(P9).


*“The poor [doctor] doesn’t even know where this is coming from… he doesn’t know how to treat him…”*
(P29).

Pediatricians were one of the specialists most frequently consulted by parents. They continued to encounter difficulties due to the lack of knowledge and management of children with PMS, but one aspect that was positively valued was the possibility of frequent visits and the possibility of seeking help for symptom control:


*“In the end, the specialists see him once a year, his pediatrician sees him almost once a month. He’s like that person we have as a reference, who reassures you because he knows who my child is and what’s wrong with him or what he needs.”*
(P25).

Parents described their experience with the emergency department as difficult due to the perceived lack of sensitivity towards their children:


*“In the emergency room, they don’t care. What they care about is recording and spelling everything right. Their biggest concern is to write the name of the syndrome correctly, nothing else is relevant…look at my son, he’ s not a syndrome.”*
(P9).

Parents described a great lack of knowledge regarding the syndrome among emergency professionals. This meant that the health professionals often doubted the symptoms reported by the parents. Because of this, most parents stated that their child had ASD, as it is a better-known condition, and treatment for the symptoms was expedited:


*“They have no idea what the syndrome is, they make funny faces… The last few times we’ve gone, I’ve gone ahead and said they have autism, so they understand and attend to it sooner and don’t leave you waiting.”*
(P22).

Previous studies [7,9] with children with an RD described the lack of knowledge of health professionals regarding the diagnosis of the RD and a decrease in their therapeutic involvement with the children due to the lack of a clear and well-determined treatment option. This led to the parents’ reluctance to go to the doctor and mistrust of health specialists [7,9]. Gómez-Zúñiga et al. [33] highlighted how trust between parents and physicians of children with an RD, and the empathy and sensitivity of the professional, are the first steps in the therapeutic relationship. Finally, Babac et al. [34] and Ivarsson et al. [35] point out the need to train healthcare professionals in RD and in the treatment and support of families.

Previous studies in hospital emergency departments point to the efficacy of family-centered care in supporting parents of children with ASD, as well as the knowledge of the pathology by healthcare personnel [36]. Our parents pointed out that in order to receive better care or not to delay their child’s care at the emergency department, they omitted the diagnosis of PMS and opted to say that their child had ASD. The researchers of the present study believe that the use of another pathology more familiar to parents is a strategy that may be useful to solve specific problems, although the key lies in the knowledge of PMS within the emergency department. Greater familiarity with this syndrome, as with ASD, would help to improve the care received by parents of children with PMS. Nicholas et al. [36] described how improving the knowledge of emergency department staff regarding ASD and focusing efforts on the family increased the efficacy of treatment in children with ASD.

### 3.5. Theme 5: Reconnection through the PMS Association

Despite social distancing, the parents described how they reconnected with others thanks to the association of parents of children with PMS. The first time they contacted the association and visited it, it was very difficult, as they met other parents and children undergoing the same reality. Nevertheless, it helped them to begin to accept it:


*“… it was super hard to get to know the association because we got a reality check. We saw how there are many children who are 100% dependent.”*
(P6).

The association is the reference center for seeking information and receiving support. Parents recounted how the role of the association is essential for filtering and confirming information about the disease:


*“… thanks to the association we have a lot of information that otherwise would not have reached us because the doctors don’t explain it to you… if we didn’t have the support of the association, we would be hopeless.”*
(P24).

In addition, the association helped them meet other families in similar circumstances, establish friendships and bonds of mutual support:


*“With them [other families] you don’t have to explain anything, and you have an amazing connection, without having anything in common other than your child’s lack of chromosomes…”*
(P4).

However, it is necessary for the parents to integrate into the association gradually because the risk of rushing into the association is that you receive a large amount of information and testimonies from other families without any filter or control, causing greater stress and anguish in parents who come to the association for the first time:


*“… I get very overwhelmed when all the parents discuss their problems. Because of that, I don’t participate much, I’m not prepared.”*
(P16).

Moreover, despite the impairment in social relationships experienced by parents of children with an RD [6,29], the associations provide significant support, and lead to friendships among families of children diagnosed with the same syndrome thanks to the understanding of their daily experience [37]. In our study, parents highlighted the role of the association as a support element and information filter. Giusti et al. [38] describe how the assistance that families of children with an RD receive through patient associations is a source of support that improves quality of life and is considered as being essential.

### 3.6. Limitations

Due to the nature of the qualitative design, the extrapolation of these results to the population of children with PMS is limited. Nevertheless, in the absence of previous studies, our results may help professionals in the educational and health care settings to understand parents of children with PMS and the daily challenges they face. The use of field notes in online chats is useful and common, but also can be biased. To decreased possible bias in the management of the field notes the authors of the present study followed the same guide to make and built researcher´s field notes (Table 2). Finally, although Excel was used for the coding and organization of the data, the use of computer-assisted qualitative data analysis (CAQDAS) using specific software is also recommended for qualitative analysis [26].

## 4. Conclusions

Our findings show how parents with children with PMS face multiple challenges in their social relationships, as a couple, within the family and their immediate social environment, and in relationships with the educational and healthcare environment.

These results show the need to: (a) create specific training programs on RD and PMS in the educational and healthcare settings; (b) promote the participation of nurses and educational professionals in the PMS association; (c) develop comprehensive programs that include educational, healthcare and social aspects in the care of children with PMS and RD, and (d) create specific care programs for parents of children with PMS, where their physical, psychological and social health are evaluated and cared for.

## Figures and Tables

**Figure 1 ijerph-19-10524-f001:**
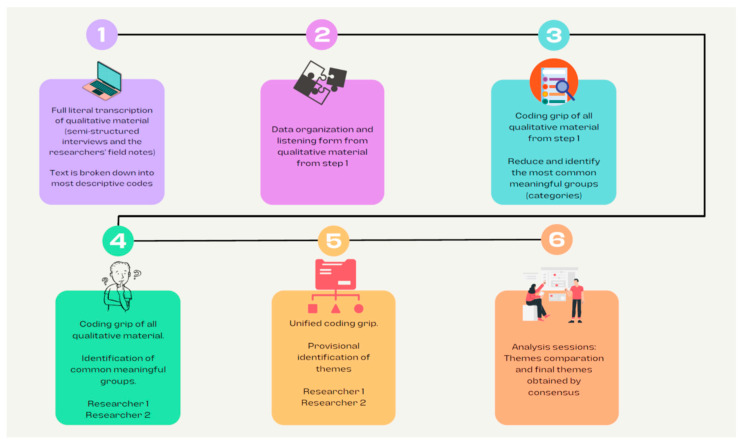
Integration and analysis process of qualitative materials.

**Table 1 ijerph-19-10524-t001:** Semi-structured question guide.

Research Areas	Questions
Illness	What is it like living with a child with Phelan McDermid Syndrome? What is most relevant to you?
Family planning	How does the disease affect your life as a couple? How does the disease affect your sexual relations? Have you considered the idea of having children again?
Family relationships	How does the disease affect your family life? What do you think about the impact or repercussion of the disease on some members of your family? Why?
Social relations	Do you think having a child with Phelan McDermid syndrome affects their social relationships? How?
Relationship with the health professional and access to health resources	How is the care you receive from the health professional? Do you think they understand your child’s illness? What is most important to you in your relationship with healthcare professionals? How do you think your relationship with your healthcare professional can influence you, your child, and your illness? Have you ever had to go to the emergency department for any reason? What was the care related to the disease like? Was it known? What would you say was most relevant about your visit to the emergency department?
Relationship with the school	At school, how would you describe the attention you receive? How is your relationship with the schoolteachers? Do you think this relationship is important?

**Table 2 ijerph-19-10524-t002:** Researcher´s field notes guide.

Areas of Field Notes	Contents of Field Notes
Content description and data collection process	Setting and context description: place of data collection. Participants: gestures, non-verbal communication, emotional responses. Content of interview. Emphasis on specific aspects narrated. Key words identified (from participant during interviews) oriented to study new concepts, contexts, situations. Incidents or relevant events during the interview: abandonment, presence of other relatives, interruption of the interview, emotional response to the questions, etc. New topics or research areas: concepts, definitions, models, theories. New bibliography to be consulted. Descriptions of the data collection process: difficulties, participant response to the use of online interviewing.
Reflexivity	Researchers´ role during data collection Researchers’ role during interview Researchers’ interpretations regarding interviews Identification of prejudices and/or stereotypes during data collection. Data collection process.

**Table 3 ijerph-19-10524-t003:** Trustworthiness criteria.

Criteria	Techniques Performed and Application Procedures
Credibility	Investigator triangulation: each interview was analyzed by two researchers. Team meetings were performed in which the analyses were compared, and categories and themes were identified.
	Triangulation of methods of data collection: semistructured interviews were conducted and researcher field notes were kept.
	Participant validation (member-checking): asking the participants to confirm the data obtained at the stages of data collection. All participants were offered the opportunity to review the audio and/or video records to confirm their experience. None of the participants made additional comments.
Transferability	In-depth descriptions of the study performed, providing details of the characteristics of researchers, participants, contexts, sampling strategies, and the data collection and analysis procedures.
Dependability	Audit by an external researcher: an external researcher assessed the research protocol, focusing on aspects concerning the methods applied and study design. An external researcher specifically checked the description of the coding tree, the major themes, participants’ quotations, quotation identification, and theme descriptions.
Confirmability	Investigator triangulation, member-checking, and data collection triangulation.
	Researcher reflexivity was encouraged by the performance of reflexive reports and by describing the rationale behind the study.

## Data Availability

The database can be accessed on request from the corresponding author.

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
