# Peer review of "Social and Family Challenges of Having a Child Diagnosed with Phelan-McDermid Syndrome: A Qualitative Study of Parents’ Experiences"

_ijerph, 2022, doi:10.3390/ijerph191710524_

Round 1
Reviewer 1 Report
An excellent paper, very well written, with a clear description of the problem and the methodology. I would like to congratulate the authors and make a few comments trying to help in the improvement of the article:
- In the Abstract, as Conclusion maybe it´s more interesting to add any of the practical proposals of the authors for the families of PMS children.
- Lines 51 to 55 are absolutely unnecessary and don´t change the paper if you delete them.
- In Method, the authors explain very well the process, but there is a small contradiction with Table 2, when about the criteria Dependability the authors include "Audit by an external researcher". Please, explain the participation of this external researcher in the whole process in the methods part.
- The aims of the study are different in the final part of the Introduction and the description in Methods. Please, clarify.
- The authors use the expressions "In relation" and "In addition..." so many times. Please, use other synonyms.
- The use of field notes in online chats is useful and common, but also can be biased and limited. Please, explain how the authors have decreased possible bias in the management of the field notes.
- Lines 123-126 are a little bit confusing, ("Additionally, during the semi-structured interviews, 32 field notes were also collected by the researchers since field notes provide a rich source of information as participants describe their personal experiences, their behaviors during data collection, and enable to note their reflections concerning methodological aspects of the data collection [28]. "). Please write them again in the clearest sense.
- The use of Excel is unusual but correct. Have the authors used any kind of formula to confirm the consensus between researchers in the selection of categories?
Author Response
RESPONSE LETTER ijerph-1850170
International Journal of Environmental Research and Public Health - Decision on Manuscript ID ijerph-1850170
Entitled: " Social and family challenges of having a child diagnosed with Phelan-Mcdermid Syndrome: A Qualitative Study of parents' experiences."
Journal: International Journal of Environmental Research and Public Health.
We would like to thank the Editors and the Reviewers for their careful consideration of our manuscript. We would also like to thank the Reviewers for their suggestions, which we believe have enhanced the quality of the manuscript. We have highlighted (in yellow) all the changes we have made throughout the text. Below, please find a detailed list of how we have addressed each comment.
REVIEWER 1
Comments and Suggestions for Authors
An excellent paper, very well written, with a clear description of the problem and the methodology. I would like to congratulate the authors and make a few comments trying to help in the improvement of the article:
- In the Abstract, as Conclusion maybe it´s more interesting to add any of the practical proposals of the authors for the families of PMS children.
ResponseThank you for the suggestion. We included at abstract section:
It would be beneficial to for parents to create training programs on PMS in the educational and healthcare settings, to promote the participation of professionals in the PMS association and to develop care programs focusing in their physical, psychological and social health.
- Lines 51 to 55 are absolutely unnecessary and don´t change the paper if you delete them.
Response: We agree with you. We have followed the reviewer´s suggestion.
- In Method, the authors explain very well the process, but there is a small contradiction with Table 2, when about the criteria Dependability the authors include "Audit by an external researcher". Please, explain the participation of this external researcher in the whole process in the methods part.
Response:
We apologize for not included it in the main text. We thought that it would be enough to present the information in the Table. We included this information alongside the methods section:
“ The study protocol was audited by an external researcher.”
Finally, the external researcher, who audited the study protocol, checked the description of the coding tree, the major themes, participants’ quotations, quotations’ identification, and themes’ descriptions.
Regarding “dependability”, Lincoln and Guba's Evaluative Criteria, reported that: “Dependability - showing that the findings are consistent and could be repeated” (available at: http://www.qualres.org/HomeLinc-3684.html)
One of the techniques used to establishing dependability is inquiry audit or External Audits. External audits as defined: “External audits involve having a researcher not involved in the research process examine both the process and product of the research study. The purpose is to evaluate the accuracy and evaluate whether or not the findings, interpretations and conclusions are supported by the data.” (available at http://www.qualres.org/HomeExte-3704.html)
For this reason an external audit was integrated to control dependability criteria. In the present study an external researcher assessed the research protocol, focusing on aspects concerning the methods applied and study design. Also, an external researcher specifically checked the description of the coding tree, the major themes, participants’ quotations, quotations’ identification, and themes’ descriptions.
- The aims of the study are different in the final part of the Introduction and the description in Methods. Please, clarify.
Response: The objective of the present study is different from the purpose of a qualitative descriptive exploratory study design (considering a generic definition of this specific design). The objective of the present study is described in the final part of the introduction. Also, the purpose of the qualitative design chosen is described at methods section.
On the one hand, the objective of the present study is described (“… to explore the experience of parents of children diagnosed with PMS in terms of interaction with their social environment.”), and on the other hand, the purpose of the qualitative descriptive exploratory study design chosen to develop the present study is described (“The purpose of an explorative descriptive qualitative study is to identify an event or a critical situation. It seeks to show "what is happening" and "how it is happening" [19,20]. Qualitative exploratory studies aim to be a comprehensive summary of events in the everyday terms of the described event. This design is the method of choice when straight descriptions of phenomena are desired [17,18].”).
We change verbs used to describe the “objective” of the present study, and the “purpose” of the design chosen.
- The authors use the expressions "In relation" and "In addition..." so many times. Please, use other synonyms.
Response: We have followed the reviewer´s suggestion.
- The use of field notes in online chats is useful and common, but also can be biased and limited. Please, explain how the authors have decreased possible bias in the management of the field notes.
Response: We have followed the reviewer´s suggestion. We included new information at data collection section (new table 2), and at limitation section (discussion).
We included new table 2 at data collection section:
|
Areas of field´s note |
Contents of field´s notes |
|
Content description and data collection process |
Setting and context description: Place of data collection. Participants: Gestures, non-verbal communication, emotional responses. Content of interview: Emphasis on specific aspects narrated. Key words identified (from participant during interviews): Oriented to study new concepts, contexts, situations. Incidents or relevant events during the interview: Abandonment, presence of other relatives, interruption of the interview, emotional response to the questions, etc. New topics or research areas: Concepts, definitions, models, theories. New bibliography to be consulted. Descriptions of the data collection process: Difficulties, participant response to the use of online interviewing. |
|
Reflexivity |
Researchers´ role during data collection Researchers´role during interview Researchers´interpretations regarding interviews Identification of prejudices and/or stereotypes during data collection. Data collection process. |
We included at limitations section:
The use of field notes in online chats is useful and common, but also can be biased. To decreased possible bias in the management of the filed notes, the authors of the present study have followed the same guide to make and built researcher´s field notes. See table 2.
- Lines 123-126 are a little bit confusing, ("Additionally, during the semi-structured interviews, 32 field notes were also collected by the researchers since field notes provide a rich source of information as participants describe their personal experiences, their behaviors during data collection, and enable to note their reflections concerning methodological aspects of the data collection [28]. "). Please write them again in the clearest sense.
Response: We agree with reviewer. We rewrite paragraph.
We included:
Additionally, during the semi-structured interviews, 32 field notes were also collected by the researchers. Researchers´ field notes provide a rich source of information as well as how participants describe their personal experiences, participants´ behaviors during data collection, and enable to researchers to describe their reflections concerning methodological aspects of the data collection [26]. See table 2.
- The use of Excel is unusual but correct. Have the authors used any kind of formula to confirm the consensus between researchers in the selection of categories?
Response: The authors used the Excel program to carry out the analysis. An in-depth knowledge of Excel allows us to make grids, with hyperlinks that allow us to organize all the codes, based on their categories, as well as to associate them (link them) to the fragments of texts from the interviews that justify their existence. In this way, information on codes, categories, themes, participants, etc. can be linked and retrieved quickly, just like software, with the advantage that it is cheaper and more accessible for teams of researchers who have economic restrictions. Moreover, the analysis is still the researcher's, not the qualitative software's. Qualitative software helps to record the data. Qualitative software helps to record, organize and retrieve information, but does not perform the analysis.
Klem et al., (20022) regarding qualitative data-management software, such as NVivo and MAXQDA, reported that: “Data-management software can be a useful adjunct to the research process and is popular among many qualitative researchers. The software is not doing the analysis; rather, consider the software as a sophisticated filing system for qualitative data.” (page 9-10)
Also, Moser & Korstjens, (2018) showed that: “Analysis software can support you to manage your data, for example by helping to store, annotate and retrieve texts, to locate words, phrases and segments of data, to name and label, to sort and organize, to identify data units, to prepare diagrams and to extract quotes. Still, as a researcher you would do the analytical work by looking at what is in the data, and making decisions about assigning codes, and identifying categories, concepts and patterns.”(page 17)
Furthermore, the international guidelines for qualitative research Consolidated criteria for reporting qualitative research (COREQ) and Standards for reporting qualitative research (SRQR) do not state that their use is mandatory.
For example, the COREQ (Tong et al., 2007) reported regarding software in the analysis and findings section: “What software, if applicable, was used to manage the data? (page 352)”. Yes, it is applicable but it does not state that it is mandatory.
Nevertheless, we agree with the reviewer that more information should be provided on this point.
New text has been included in the analysis section:
No qualitative software was used to analyze the data. For the analysis, the Excel program was used to organize and share the coding process. Excel allow to make grids, with hyperlinks that allow us to organize all the codes, based on their categories, as well as to associate them (link them) to the fragments of texts from the interviews that justify their existence.
Also, edits have been made to the limitations section:
Furthermore, although Excel was used for the coding and organization of the data, the use of the computer assisted qualitative data analysis (CAQDAS) using specific software is also recommended for the qualitative analysis [28].
On the other hand, the authors did not use any kind of formula to confirm the consensus between researchers in the selection of categories.
The authors used an inductive thematic analysis, they did not determine a list of categories before analysis to help or could calculate any type of statistical concordance.
The consensus process among researchers consist on each researcher (researcher 1 and 2) performed a double, and independent coding in each interview and each field note. Subsequently, both researchers met to discuss, compare, and refine their categories and themes. Comparation process included comparation of categories definition´s, their characteristics and narratives which justified all categories obtained. After that, the same process occurred with themes. Additionally, joint meetings were held to combine the results of the analysis. In the case of differences in opinion, theme identification was performed based on establishing a consensus.
Nevertheless, we included new information at analysis section:
Subsequently, they met to discuss, compare, and refine their findings. Comparation and refine process included study and evaluation of categories definition´s, its characteristics and narratives which justified all categories obtained. After that, the same process occurred with themes. Additionally, joint meetings were held to combine the results of the analysis, to represent the parents’ experiences [21,26]. In the case of differences in opinion, theme identification was performed based on establishing a consensus.
References:
- Klem, N. R., Shields, N., Smith, A., & Bunzli, S. (2022). Demystifying Qualitative Research for Musculoskeletal Practitioners Part 4: A Qualitative Researcher's Toolkit-Sampling, Data Collection Methods, and Data Analysis. The Journal of orthopaedic and sports physical therapy, 52(1), 8–10. https://doi.org/10.2519/jospt.2022.10486
- Moser, A., & Korstjens, I. (2018). Series: Practical guidance to qualitative research. Part 3: Sampling, data collection and analysis. The European journal of general practice, 24(1), 9–18. https://doi.org/10.1080/13814788.2017.1375091
- O'Brien, B. C., Harris, I. B., Beckman, T. J., Reed, D. A., & Cook, D. A. (2014). Standards for reporting qualitative research: a synthesis of recommendations. Academic medicine : journal of the Association of American Medical Colleges, 89(9), 1245–1251. https://doi.org/10.1097/ACM.0000000000000388
- Tong, A., Sainsbury, P., & Craig, J. (2007). Consolidated criteria for reporting qualitative research (COREQ): a 32-item checklist for interviews and focus groups. International journal for quality in health care : journal of the International Society for Quality in Health Care, 19(6), 349–357. https://doi.org/10.1093/intqhc/mzm042
We hope that you are satisfied with the revision and that this manuscript may now be suitable for publication in International Journal of Environmental Research and Public Health.
Sincerely,
The Authors

Reviewer 2 Report
First of all, congratulations on your work and the inherent quality.
Overall I found your article very interesting, with an excellent flow throughout the text. The research problem is clear and well contextualized. The methods are well described and adequate to answer the research problem. Results are well presented and help the reader understand the themes that emerged from the analysis.
My only recommendation (not mandatory) is that the discussion could benefit from a clearer integration with the results. Could integrating the results with the discussion be possible? This way, there would be a better flow. Another possibility would be to do a small contextualization of the results before the confrontation with the literature.
Just a small correction in line 106, replace "In in" with "In".
Once again, congratulations on your work.
Author Response
RESPONSE LETTER ijerph-1850170
International Journal of Environmental Research and Public Health - Decision on Manuscript ID ijerph-1850170
Entitled: " Social and family challenges of having a child diagnosed with Phelan-Mcdermid Syndrome: A Qualitative Study of parents' experiences."
Journal: International Journal of Environmental Research and Public Health.
We would like to thank the Editors and the Reviewers for their careful consideration of our manuscript. We would also like to thank the Reviewers for their suggestions, which we believe have enhanced the quality of the manuscript. We have highlighted (in yellow) all the changes we have made throughout the text. Below, please find a detailed list of how we have addressed each comment.
REVIEWER 2
Comments and Suggestions for Authors
First of all, congratulations on your work and the inherent quality. Overall I found your article very interesting, with an excellent flow throughout the text. The research problem is clear and well contextualized. The methods are well described and adequate to answer the research problem. Results are well presented and help the reader understand the themes that emerged from the analysis.
My only recommendation (not mandatory) is that the discussion could benefit from a clearer integration with the results. Could integrating the results with the discussion be possible? This way, there would be a better flow. Another possibility would be to do a small contextualization of the results before the confrontation with the literature.
Response: We agree with reviewer. We have followed the reviewer´s suggestion. We integrate the results with discussion. We create a new “Results and discussions” section.
Just a small correction in line 106, replace "In in" with "In".
Response: We have followed the reviewer´s suggestions.
We hope that you are satisfied with the revision and that this manuscript may now be suitable for publication in International Journal of Environmental Research and Public Health.
Sincerely,
The Authors

Reviewer 3 Report
Well-presented manuscript, attempting to present a qualitative analysis of parents' experiences chaving a child diagnosed with Phelan-Mcdermid Syndrome. Nevertheless, the novelty and scientific soundness of your results is restricted.
Author Response
RESPONSE LETTER ijerph-1850170
International Journal of Environmental Research and Public Health - Decision on Manuscript ID ijerph-1850170
Entitled: " Social and family challenges of having a child diagnosed with Phelan-Mcdermid Syndrome: A Qualitative Study of parents' experiences."
Journal: International Journal of Environmental Research and Public Health.
We would like to thank the Editors and the Reviewers for their careful consideration of our manuscript. We would also like to thank the Reviewers for their suggestions, which we believe have enhanced the quality of the manuscript. We have highlighted (in yellow) all the changes we have made throughout the text. Below, please find a detailed list of how we have addressed each comment.
REVIEWER 3
Comments and Suggestions for Authors
Well-presented manuscript, attempting to present a qualitative analysis of parents' experiences chaving a child diagnosed with Phelan-Mcdermid Syndrome. Nevertheless, the novelty and scientific soundness of your results is restricted.
Response: Thank you for your comments. Also, we revised published bibliography regarding Phelan-Mcdermid Syndrome and qualitative researchs and parents (at pub med) (https://pubmed.ncbi.nlm.nih.gov/?term=%28Phelan-Mcdermid+Syndrome+AND+qualitative+researchs+AND+parents%29&sort=pubdate&size=100) and we did not obtain results. We believe that the present study is the first to describe the experience of parents of children diagnosed with PMS in terms of interaction with their social environment.
We hope that you are satisfied with the revision and that this manuscript may now be suitable for publication in International Journal of Environmental Research and Public Health.
Sincerely,
The Authors

Round 2
Reviewer 3 Report
Nicely presented manuscript with significant corrections and improvements relative to the previous version that was submitted for review. I suggest that this work could be concidered for publication to this Journal.